# Perturbation of *TM6SF2* Expression Alters Lipid Metabolism in a Human Liver Cell Line

**DOI:** 10.3390/ijms22189758

**Published:** 2021-09-09

**Authors:** Asmita Pant, Yue Chen, Annapurna Kuppa, Xiaomeng Du, Brian D. Halligan, Elizabeth K. Speliotes

**Affiliations:** 1Division of Gastroenterology and Hepatology, University of Michigan Health System, Ann Arbor, MI 48109, USA; pantas@med.umich.edu (A.P.); yuech@med.umich.edu (Y.C.); kuppaa@med.umich.edu (A.K.); xmdu@med.umich.edu (X.D.); halligan@med.umich.edu (B.D.H.); 2Department of Computational Medicine and Bioinformatics, University of Michigan Medical School, Ann Arbor, MI 48109, USA

**Keywords:** transmembrane 6 superfamily member 2, non-alcoholic fatty liver disease, triglycerides, lipidomics, RNASeq

## Abstract

Non-alcoholic fatty liver disease (NAFLD) is caused by excess lipid accumulation in hepatocytes. Genome-wide association studies have identified a strong association of NAFLD with non-synonymous E167K amino acid mutation in the transmembrane 6 superfamily member 2 (TM6SF2) protein. The E167K mutation reduces TM6SF2 stability, and its carriers display increased hepatic lipids and lower serum triglycerides. However, the effects of TM6SF2 on hepatic lipid metabolism are not completely understood. We overexpressed wild-type or E167K variant of *TM6SF2* or knocked down *TM6SF2* expression in lipid-treated Huh-7 cells and used untargeted lipidomic analysis, RNAseq transcriptome analysis, and fluorescent imaging to determine changes in hepatic lipid metabolism. Both *TM6SF2* knockdown and E167K overexpression increased hepatic lipid accumulation, while wild-type overexpression decreased acylglyceride levels. We also observed lipid chain remodeling for acylglycerides by *TM6SF2* knockdown, leading to a relative increase in species with shorter, more saturated side chains. RNA-sequencing revealed differential expression of several lipid metabolizing genes, including genes belonging to AKR1 family and lipases, primarily in cells with *TM6SF2* knockdown. Taken together, our data show that overexpression of *TM6SF2* gene or its loss-of-function changes hepatic lipid species composition and expression of lipid metabolizing genes. Additionally, our data further confirms a loss-of-function effect for the E167K variant.

## 1. Introduction

Non-alcoholic fatty liver disease (NAFLD) is one of the most common chronic liver diseases in the world [1]. NAFLD pathology is variable and can range from simple steatosis to steatohepatitis and cirrhosis [2]. No effective treatments exist for NAFLD, making it a large unmet medical need. A better understanding of causes of NAFLD is important for developing novel therapeutic strategies for this disease. NAFLD is heritable and polygenic as evident by a growing list of NAFLD-related single nucleotide polymorphic risk variants identified from genome-wide association studies (GWAS) [3]. One NAFLD associated variant, rs58542926, falls in the transmembrane 6 superfamily member 2 (*TM6SF2*) gene and causes a nonsynonymous glutamic acid to lysine substitution at the amino acid residue 167 (E167K) [4]. Presence of the E167K variant in humans is associated with higher hepatic lipid content and has been shown to be a strong genetic risk factor for NAFLD, fibrosis and cirrhosis [4,5,6,7,8].

In vitro studies suggest that *TM6SF2* E167K variant results in significantly reduced expression of the TM6SF2 protein in liver [4,9]. Knocking out *Tm6sf2* in mice or inhibition of *TM6SF2* gene expression in hepatocytes recapitulates the human phenotype of E167K carriers, suggesting that the *TM6SF2* E167K variant acts by a loss-of-function mechanism [4,10]. The gene is predominantly expressed in the liver and small intestines and is mainly localized to the endoplasmic reticulum (ER) and the ER-Golgi intermediate compartment, where triglyceride (TG) rich lipoproteins (TRLs) are assembled [4,10]. TM6SF2 may have a role in the assembly of very-low density lipoproteins (VLDL) and studies have reported impaired lipidation of the VLDL particles and lower levels of plasma VLDL-TGs in human E167K carriers and in mice with *Tm6sf2* knockout [11,12,13,14,15]. The mechanism by which E167K or the loss of TM6SF2 function causes hepatic steatosis, steatohepatitis, and fibrosis is less understood. While spontaneous increase in hepatic TGs under basal conditions is one of the main phenotypes associated with the *TM6SF2* variant, how the increased or decreased function of TM6SF2 protein affects lipid droplet properties, expression of lipid metabolizing genes or cellular lipid composition is not completely understood. Indeed, many argue that a neutral lipid like triglyceride is likely not the cause of NAFLD-related liver damage and such damage may be caused by the accumulation of more hepatotoxic lipids such as saturated fatty acids, diacylglycerols, ceramides, lysophophatidyl choline, phosphoinositol, and free cholesterols, which may also accumulate in NAFLD, albeit at lower amounts than TG [16,17].

Here, our objective is to explore how changes in *TM6SF2* expression affect lipid metabolism in hepatic cells under lipid exposure. For this, we overexpressed wild-type or E167K variant forms of *TM6SF2* or knocked down *TM6SF2* in oleic acid-treated Huh-7 cells. We used biochemical assays, untargeted lipidomic analysis, RNAseq transcriptome analysis and high-throughput fluorescent microscopic image analysis to determine the cellular changes associated with altering TM6SF2 protein.

## 2. Results

### 2.1. Effects of TM6SF2 Expression on Lipid Profile in Oleic Acid-Treated Huh-7 Cells

In order to investigate the cellular changes resulting from the overexpression or the knockdown of *TM6SF2* in lipid-treated Huh-7 cells, we first generated Huh-7 cell lines with stable overexpression of wild-type or the E167K mutant forms of *TM6SF2* through a strong pCMV6 promoter and a stable *TM6SF2* knockdown cell line using TRC pLKO1.0 shRNA lentiviral clones. As controls, we also constructed cell lines with an integrated lentivirus expressing a non-targeting shRNA and or an empty pCMV6-Entry vector. *TM6SF2* gene expression was increased ~10-fold and 8-fold for wild-type or E167K variant overexpression, respectively (Appendix A) whereas *TM6SF2* mRNA levels were reduced by 75% with *TM6SF2* knockdown (Appendix A). We then determined how changes in *TM6SF2* expression affects lipid accumulation in oleic acid-treated Huh-7 cells. We first lipid starved our control and experimental cells to achieve a low lipid background and then treated them with 200 µM oleic acid for 24 h. Overexpression of the E167K variant as well as knockdown of *TM6SF2* both increased the total intracellular acylglyceride (TG, DG, and MG) levels by ~1.8-fold, whereas overexpression of wild-type lowered the acylglyceride levels by 52% (Figure 1A,E).

Next, we examined the properties of lipid droplets in lipid-loaded Huh-7 cells using high-content imaging and an automated image analysis pipeline (detail protocol in the Section 4). Briefly, we used Hoechst and LipidTox green dyes to stain the nuclei and lipids, respectively. We then applied a CellProfiler analysis pipeline for automated identification of lipid droplets, nuclei and cell boundary (Appendix A). We quantitated multiple lipid droplet associated variables in our cells, including mean intensity and mean area of individual lipid droplets, as well as the mean number of lipid droplets per cell. The imaging and CellProfiler analysis pipeline was first validated in our vector control cell lines that were treated with an increasing dose of oleic acid (0, 200, and 1000 µM) for 24 h where we observed a dose-dependent increase in the mean intensity and mean area of lipid droplets, and mean number of lipid droplets per cell (Appendix A). We then applied the same analysis pipeline to compare lipid droplet variables between our control and overexpression or knockdown cell lines that were treated with 200 or 1000 µM oleic-acid for 24 h. Mean intensity of lipid droplets in cells with E167K variant overexpression was significantly higher at both 200 or 1000 µM oleic acid concentrations (Figure 1B) and in *TM6SF2* knockdown cells at 1000 µM oleic acid concentrations (Figure 1F) compared to their respective controls at the same concentrations of oleic acid. Similarly, the mean area of lipid droplets was significantly increased by E167K variant overexpression at 200 and 1000 µM oleic acid concentrations (Figure 1D), while neither wild-type overexpression or knockdown of *TM6SF2* had any effect on mean lipid droplet area (Figure 1D,H). We did not observe any significant changes for lipid droplet number per cells in our overexpression or knockdown cells when compared to their respective control cells across all treatment conditions (Figure 1C,G). Compared to wild-type overexpression, E167K overexpression resulted in lipid droplets with significantly higher mean intensity at both oleic acid concentrations (Figure 1B) and mean droplet area at 1000 µM (Figure 1D) while number of lipid droplets per cell was not significantly different (Figure 1C). We observed smaller magnitude of changes for our lipid variables in imaging assays compared to absolute acylglyceride amounts determined biochemically (Appendix A). This suggests that no single variable measured in imaging assays perfectly mirrors the total change in acylglyceride levels biochemically. The imaging however allows us to quantify cellular features (lipid droplet size, number and intensity) that are not discernable from a bulk biochemical assay that in some cases may have biological significance. It is possible that combinations of the imaging variables may better correlate with the biochemical measures. Congruent results from our biochemical and image assays suggest that *TM6SF2* knockdown and E167K variant overexpression increase acylglyceride accumulation and mean intensity of lipid droplets in oleic acid treated Huh-7 cells. As the E167K variant is considered to be a loss of function mutation, its overexpression may cause non-physiological effects. For further experiments, we focused on characterizing the effects of loss or overexpression of wild-type *TM6SF2* function.

### 2.2. Differential Effects of TM6SF2 Genotypes on Hepatic Lipid Profile

Although accumulation of TGs is the major phenotype in E167K carriers or when *TM6SF2* is knocked down, neutral lipids like TGs may not cause NAFLD-associated liver damage, but rather the accumulation of other more toxic lipids may determine severity and progression of NAFLD. To determine what lipid species were altered when *TM6SF2* was perturbed, we carried out untargeted lipidomic analysis by liquid chromatography coupled to high-resolution mass spectrometry (LC-MS/MS). Approximately 350 different lipid species distributed across 17 classes were characterized. Unsupervised hierarchical clustering of lipid species across all samples showed that replicate experimental and control samples clustered as groups, showing good concordance between replicate runs (Appendix A). Both knockdown of *TM6SF2* and wild-type overexpression significantly (adjusted *p* value < 0.05 and log fold change > 1 or <−1) changed the abundance of 30 and 27 lipid species, respectively, when compared to their control cells (Appendix A). Most of the changes were observed across lipid classes of TGs, DGs, phospholipids (PC, PE, PS, and PG), sphingomyelins, lyso phospholipids and plasmenyl phospholipids, and the detail changes are shown in Appendix A.

Wild-type overexpression significantly changed the abundance of 27 lipid species when compared to the vector control (Appendix A and Figure 2A). The two lipid species with the largest log_2_ fold change were plasmenylPC 44:4, which increased by approximately 11 fold with wild-type overexpression and plasmenylPE 34:0, which decreased by 3.4 fold. The two lipid species with the smallest adjusted *p* value were PE 31:1 and PE 32:2 (adjusted *p* value of 0.002), which both increased by 2.8 and 3.4 fold, respectively, with wild-type overexpression. Two triglyceride species, TG 52:6 and TG 60:5, were both decreased by approximately 2.7 fold, and three diglyceride species, DG 36:3, DG 38:2, and DG 38:4, were also decreased by approximately 2.4 fold by wild-type overexpression. In the knockdown comparison (Figure 2B), the highest log_2_ fold change was found with lysoPE 20:3, which increased by approximately 4 fold with knockdown and PC 42:9, which decreased approximately 4.6 fold with knockdown. Five lipid species with the lowest adjusted *p* value (7 × 10^−5^) were lysoPC 18:0, lysoPC 20:3, lysoPE 20:3, PG 38:7, and plasmenylPE 38:3, all of which increased in the knockdown by 2 to 3 fold. Four other lipid species with the lowest adjusted *p* value (7 × 10^−5^) were SM 33:0, TG 58:8, TG 60:7, and TG 62:7, all of which decreased in the knockdown by approximately 2.1 fold. There were 7 triglyceride species that were significantly changed between the knockdown and non-targeted control cells. In the knockdown, TG 52:1 and TG 58:1 were increased by approximately 2 fold and TG 56:1, TG 58:7, TG 58:8, TG 60:7, and TG 62:7 were decreased by approximately 2 fold. The two diglycerides, DG 31:0 and DG 34:0, were increased by 2.1 and 2.3 fold in the knockdown.

We also analyzed lipidomic results for changes in side chain length and unsaturation across observed lipid classes. Heat maps showing the distribution of unsaturation and chain length for each lipid class are shown in Appendix A. Overall, there were small changes in the distribution of chain length and saturation for most lipid classes. The most significant changes were observed for DGs and TGs (Figure 3). Wild-type *TM6SF2* overexpression generally decreased the abundance of TG and DG species across all saturation and chain lengths (Figure 3A,C). In contrast, *TM6SF2* knockdown significantly increased the abundance of more saturated TGs and DGs with shorter side-chain length (<16 carbon per side-chain), while the abundance of TGs and DGs with higher side-chain lengths and unsaturation was decreased (Figure 3B,D). Since some of the highest levels of changes in abundance for lipid species were observed for lyso lipids, particularly the lysoPCs and lysoPEs in the *TM6SF2* knockdown cells, we also looked at the relative changes in abundance of lyso lipids by chain length and unsaturation (Appendix A). However, we did not observe any major shift in the distribution of chain saturation or side-chain length for lyso lipids, which is in contrast to acyl lipids where it seemed that the change in abundance was correlated to a shift in chain length and unsaturation instead of increase in a specific lipid species.

Taken together, our results show that knockdown of *TM6SF2* increased DGs, lysoPC, and lysoPE lipids, decreased PCs and PEs, and also caused a shift in acyl glyceride lipid side chains to become more saturated with shorter chain lengths. Wild-type TM6SF2 overexpression decreased TGs, DGs, MGs, and PGs, increased PCs, PEs, and lysoPEs, and had no effect on chain lengths of lipids.

### 2.3. RNAseq Analysis Show Changes in Expression of Genes Involved in Lipid Metabolism in Response to Different TM6SF2 Expression Conditions

Next, we performed RNAseq analysis to determine changes in gene expression resulting from altered *TM6SF2* expression. We focused our analysis on the combination of the set of 1008 genes for the combination of the 749 proteins identified by Reactome as being part of the overall human metabolism of lipids pathway (R-HSA-556833) and the 864 human proteins identified in GO as being involved in the regulation of lipid metabolic process (GO:0019216). Volcano plots of differential expression of these genes are shown in Figure 4. Compared to empty vector, 16 genes were differentially expressed (adjusted *p* value of less than 0.05 and an absolute log fold change of greater than 0.5 or 1.4-fold change) in cells with wild-type overexpression (Figure 4A), while 96 lipid metabolizing genes were differentially expressed in the cells with *TM6SF2* knockdown compared to the non-targeting control (Figure 4B). Complete sets of genes are listed in the Appendix A. Four members of the Aldo-keto reductase family (*AKR1B1, AKR1C2, AKR1C4, AKR1D1*) were increased from 2 to 6 fold by *TM6SF2* knockdown whereas expression of *AKR1C4, AKR1C2 and AKR1C1* were all decreased by at least 1.5-fold in wild-type overexpression cells. Expression of both *LDLR* (low-density lipoprotein receptor) and (*LSR*) lipolysis-stimulated lipoprotein receptor was increased by ~1.4-fold and of was decreased by ~1.8-fold. We also observed changes in the expression of apolipoproteins such as *APOA5* (Apolipoprotein A-V) and *APOC3* that decreased by ~1.8 and ~2-fold, respectively, by knockdown. In contrast, decreased expression of *APOA2* (~1.5-fold) and increased expression of *APOD* (~4-fold) were observed in cells with wild-type overexpression. Across the two comparisons, 16 different genes involved in the metabolism of lysophosphatidylethanolamine (*LPE*) were significantly changed. *LIPH*, a gene involved in LPE hydrolysis, was increased in both conditions, and *PNPLA2* was decreased in knock down of *TM6SF2*. *TM6SF2* RNA expression was increased by 3.4 fold in the wild-type overexpression and reduced to 0.82 in the knockdown, consistent with the qPCR results (Appendix A).

## 3. Discussion

The primary objective of this study was to determine the differential effects of *TM6SF2* overexpression and loss on hepatic lipid metabolism, composition and lipid droplet dynamics under steatogenic conditions. We utilized four parallel experimental approaches to determine changes in total levels of acylglycerides, gene expression, individual lipid species, and lipid droplet phenotype in oleic acid-treated Huh-7 cells. Our data shows that knockdown of *TM6SF2* and overexpression of the E167K mutant variant increases whereas overexpression of the wild-type lowers acylglycerol accumulation in lipid-loaded hepatocytes. While we did not see changes across lipid classes, we observed lipid chain remodeling for TGs and DGs by *TM6SF2* knockdown, leading to a relative increase in shorter and more saturated side chains on glycerol lipids. Importantly, *TM6SF2* knockdown and overexpression lead to significant changes in the abundance of several lipid species, including phosphoplipids, lysophospholipids, and acylglycerides. RNA sequencing revealed differential expression of lipid metabolizing genes belonging to the *AKR1* family and lipases with *TM6SF2* knockdown causing most of the significant changes. Characterization of lipid droplets profile through high-content image analysis determined that overexpression of the E167K variant and *TM6SF2* knockdown significantly increase the staining intensity and size of lipid droplets. Taken together, our data shows that overexpression of *TM6SF2* gene or its loss of function significantly and differentially affects intrahepatic lipid metabolism to change lipid species composition, expression of lipid metabolizing genes and lipid droplet profile in oleic acid- treated Huh-7 cells.

While data from multiple human studies have established the strong association of *TM6SF2* E167K mutant variant with NAFLD, the exact mechanism how the loss of TM6SF2 function causes hepatic steatosis is not entirely clear. Study in NAFLD patients and in animal models have largely focused on understanding how the E167K variant or the *TM6SF2* knockdown affects very-low density lipoprotein (VLDL) and secretion of VLDL TG, and on TGs levels in plasma or in the liver [12,13,14,15,18,19,20]. Similarly, most of the in vitro studies on *TM6SF2* knockdown or overexpression explored the effects of gene expression modification in hepatic lipid accumulation under basal conditions and in the absence of lipid exposure [10,21]. Here, we evaluated the effects of *TM6SF2* overexpression or its loss of function on intrahepatic lipid metabolism under a steatogenic environment and show that increased accumulation of intracellular lipids is clearly evident in oleic acid-treated Huh-7 cells with *TM6SF2* knockdown or E167K variant overexpression. Additionally, in the present study, we utilized high-throughput imaging with high-content image analysis to determine changes in hepatic lipid accumulation at individual lipid droplet level. Our results show significant increases in intensity and area of individual lipid droplets with *TM6SF2* knockdown or E167K variant overexpression, suggesting the presence of larger, more TG rich lipid droplets in the cells. One possible mechanism behind this observation could be the previously demonstrated role of TM6SF2 on lipidation of very low density lipoproteins and data showing that its loss of function results in intrahepatic retention of TGs [13]. Previous studies that investigated the effects of E167K variant overexpression or *TM6SF2* knockdown did so separately in animal or cell-based models and suggest a loss of function effects for the variant [4,10]. Recently, Li et al. showed that overexpression of E167K in Huh-7 cells reduces TM6SF2 protein expression compared to wild-type overexpression and act similarly to *TM6SF2* knockdown to destabilize APOB48 expression [20]. In this study, we show that E167K variant overexpression recapitulates the effects of *TM6SF2* knockdown on hepatic lipid accumulation and lipid droplets’ profile suggesting E167K protein may possibly have a dominant negative mechanism of action on hepatic lipid metabolism when overexpressed.

*TM6SF2* E167K variant has also been associated with progression of NAFLD to NASH [7,8]. Although accumulation of TGs is the major phenotype in E167K carriers, neutral lipids like TGs may not lead to NAFLD-associated liver damage, but rather the accumulation of other more toxic lipids may determine severity and progression of NAFLD. In the current study, overexpression of the wild-type *TM6SF2* did not cause any major changes in lipid species that have been previously shown to cause liver damage in NAFLD. However, *TM6SF2* knockdown significantly increased in lysoPC and lysoPE, both of which were recently identified as early biomarkers of NAFLD and are increased in NALFD patients with hepatitis [22]. LysoPC has also been previously shown to be increased in liver of NASH patients [23,24,25]. However, lysoPE, but not lysoPCs, were also increased by wild-type overexpression; therefore, further study is needed to determine the significance of these changes.

Biochemical assessment showed significant increases in the total acylglyceride levels in our cells with *TM6SF2* knockdown compared to non-target controls. Similarly, the levels of DGs were generally increased by lipidomic analysis in *TM6SF2* knockdown cells compared to non-target controls. We observed mixed (some increased and some decreased) patterns of changes for particular TG species in *TM6SF2* knockdown compared to non-target controls. Previous studies show that hepatic TGs or DGs extracted from the VLDLs in liver of E167K mutant carriers have shorter and more saturated fatty acids [15,26]. In accord with these findings, we observed relative increase in the abundance of more saturated acylglycerides with shorter side chain lengths in cells with *TM6SF2* knockdown. In addition to TGs, previous studies also suggest that *TM6SF2* knockdown may lead to a global shift in lipids in terms of saturation, including PCs, which are major membrane phospholipids [27]. At the same time, studies show variable results on *TM6SF2* knockdown-mediated changes in the total abundance of hepatic PCs with data showing both decrease or increase in Huh-7 cells, no change in mice and decrease in human E167K carriers [13,26,27]. Luukkonen et al. also reported that PCs synthesis from polyunsaturated fatty acids is impaired in E167K variant carriers thereby lowering total PCs in liver. Although we did not see total class level changes for PCs in our study, abundance of two of the PC species were significantly decreased by *TM6SF2* knockdown, while opposite effects were observed for wild-type *TM6SF2* overexpression where abundance of three out of four significantly changed PCs were increased. Similar effects were also observed for significantly changed PE species, all of which were decreased by *TM6SF2* knockdown and increased by wild-type overexpression. Changes at absolute levels of acyglycerides biochemically may not be reflective of changes in total acylglycerides measured in lipidomic analyses as the lipidomic analyses do not measure all possible acylglyceride species. Members of aldo-keto reductase 1 (AKR1) family, enzymes that regulate steroid metabolism, have been previously known to play a major role in hepatocarcinoma [28]. Increasing evidence suggest that several members of the AKR1 family such as AKR1B7 and AKR1B10 might play important role in the development of NAFLD or NASH [29,30,31]. Here, expression of several AKR1 family of enzymes were either increased or decreased in all of our cell lines compared to controls, with most changes observed for cells with *TM6SF2* knockdown. We also found differential gene expression for lipases *PNPLA2* and *LIPH* in our study along with changes in the expression of lipoprotein receptor genes (*LDLR, LSR, LRP2*), which could be cellular response to increased TG levels in the cells. Further investigation is needed to reveal any mechanistic association of these genes in *TM6SF2*-mediated changes in hepatic lipid levels.

In this study, we combined systems biology, biochemical assays and high-throughput high-content image analysis to get a more complete picture of hepatic lipid metabolism-associated changes resulting from alterations in *TM6SF2* gene expression than currently available. Our data shows some novel changes in lipid composition and expression of lipid metabolizing genes; however, the mechanism of how these changes manifest and their role, if any, on hepatic steatosis and other NAFLD-related phenotypes needs further investigation. In summary, the current study shows that overexpression of wild-type *TM6SF2* or loss of *TM6SF2* function can effectively and differentially modulate hepatic lipid metabolism to produce significant changes in intracellular lipid levels and lipid species composition. Importantly, loss of *TM6SF2* function may produce more profound and overall changes in lipid metabolic pathways under a steatogenic environment. Finally, we have shown that the in vitro model used in this study, particularly the *TM6SF2* knockdown studies, recapitulates many of the lipid-metabolism associated phenotypes that were observed in human E167K carriers. Thus, use of this in vitro model along with high-throughput experimental study design such as high-content image analysis used in this study could be a valuable tool to perform further functional analysis on *TM6SF2* or other GWAS identified genetic modifiers of hepatic steatosis.

## 4. Methods

### 4.1. Cell Line Generation

#### 4.1.1. Cloning Human *TM6SF2* cDNA into pCMV6-Entry Vector

All experiments using lentivirus were performed at the University of Michigan under Biosafety Level 2 (BSL2) protocols in compliance with containment procedures in laboratories approved for use by the University of Michigan Institutional Biosafety Committee (IBC) and Environment, Health and Safety (EHS). Huh-7 human hepatocarcinoma cell line was used for functional analyses [32].

Human wild-type *TM6SF2* was amplified by PCR from a human cDNAs library using primers containing Mlu I and EcoR I restriction sites and was sub cloned into the pCMV6-Entry vector (OriGene, Rockville, MD, USA) using the Gibson cloning kit with Q5 High-Fidelity DNA Polymerase (New England Biolabs, Ipswich, MA, USA). The sequences of *TM6SF2* insert was confirmed by Sanger sequencing with 3X coverage of sequencing (University of Michigan Sequencing Core, Ann Arbor, MI, USA).

#### 4.1.2. Preparation of Human pCMV6-*TM6SF2* (E167K) Mutant cDNA Plasmid

The coding sequence for the human *TM6SF2* mutant variant (E167K) cDNA was amplified from wild- type *TM6SF2* cDNA plasmid (pCMV6-*TM6SF2* plasmid DNA as PCR template) by PCR site-directed mutagenesis using QuikChange^TM^ Site-Directed Mutagenesis Kit (Stratagene, San Diego, CA, USA). The sequences of *TM6SF2* mutant variant (E167K) construct was confirmed by Sanger sequencing with 3X coverage of sequencing (University of Michigan Sequencing Core).

#### 4.1.3. Overexpression of *TM6SF2* (Wild-Type and E167K Mutant) Genes in Cultured Human Hepatoma Cells

Huh-7 cells were grown to 70% confluence in DMEM medium with 10% FBS plus 100 IU/mL penicillin and 100 µg/mL streptomycin. For *TM6SF2* overexpression, Huh-7 cells were transfected with the pCMV6-Entry vector expressing *TM6SF2* (wild-type or E167K mutant) genes or a control empty pCMV6-Entry vector using FuGENE transfection reagent (Thermo Fisher Scientific, Waltham, MA, USA). Post transfection (48 h), cells stably expressing these genes were selected with G-418 (10 µg/mL) for 72 h. Culture DMEM medium was freshly changed and fresh G-418 was added every 2 days. After 3 days, cells stably overexpressing the *TM6SF2* (wild-type or E167K mutant) gene were collected, and the total cellular RNA was extracted using TRIzol reagent (Thermo Fisher Scientific, Waltham, MA, USA). Overexpression was measured and confirmed by RT-qPCR.

#### 4.1.4. Knockdown of Endogenous *TM6SF2* Gene Expression in Cultured Human Hepatoma Cells

Huh-7 cells were grown to 70% confluence in DMEM medium with 10% FBS plus 100 IU/mL penicillin and 100 µg/mL streptomycin and infected with human *TM6SF2* shRNAs (clone 382425, clone 382426) lentivirus (Sigma-Aldrich Company, St. Louis, MO, USA). At 72 h post infection, Huh-7 cells stably expressing *TM6SF2* shRNAs lentivirus vector were selected in new C-DMEM medium containing 10 µg/mL puromycin (Sigma-Aldrich Company) for 7 days. The culture medium was changed, and fresh puromycin (10 µg/mL) was added every 2 days. After 7 days, Huh-7 cells with stable *TM6SF2* knock down were collected, and total cellular RNA was extracted using TRIzol reagent. Knockdown of *TM6SF2* mRNA was confirmed by RT-PCR.

### 4.2. Experimental Design

For our studies, Huh-7 cells (controls or genetically modified cells) were cultured in high glucose (25 mM) DMEM with 10% FBS and 1% PenStrep along in the presence of antibiotics. After 24 h, the growth medium was replaced with DMEM containing 10% delipidated FBS. Twenty-four hours after lipid starvation, 200 µM of BSA-conjugated oleic acid was added to DMEM culture medium. Cells were harvested after 24 h and samples were collected for RNA, protein, and intracellular triglyceride measurements and lipids were extracted for LC-MS/MS untargeted lipidomics analysis. Similarly treated cells were also fixed and stained for image analysis using high-throughput microscopy.

### 4.3. Gene Expression Analysis

*TM6SF2* gene expression was measured by real time quantitative RT-qPCR method, and total cellular RNA was prepared from stable overexpression/knockdown Huh-7 cells using TRIzol reagent (Thermo Fisher Scientific) and purified by ZYMO Clean Kit. Superscript VILO (Life Technologies, Carlsbad, CA, USA) reverse transcriptase kit was used to synthesize the first strand cDNA, and reverse transcribed cDNA served as template in polymerase chain reactions. The real-time relative quantitative PCR was performed using TaqMan^®^ Gene Expression Assay (Life Technologies) FAM probes for *TM6SF2* gene, respectively, with TagMan^®^ Gene Expression Master Mix (Life Technologies) following the manufacturer’s protocol. The TaqMan^®^ ELF1 FAM Gene Expression Assay probes (Life Technologies) were used as endogenous gene control.

### 4.4. Acylglyceride Assay

After harvesting, total cellular triglycerides were extracted from cells using the Folch extraction method and quantified by using a Serum Triglyceride Determination Kit (Sigma-Aldrich Company). The absorbance was read at 540 nm using an Eppendorf Biospectrometer and 2.5 mg/mL glycerol was used as the standard (Sigma-Aldrich Company). The total cellular triglyceride content was normalized to total cellular protein.

### 4.5. Sample Preparation for Untargeted Lipidomics

Twenty-four hours after oleic acid exposure, cell culture medium was removed and cells were washed twice with 1X PBS buffer. Cells were then collected by 0.05% Trypsin-EDTA lysis, counted using LUNA^TM^ automated cell counter (Annandale, VA, USA) and stored at −80 °C until analyzed. The lipids were extracted using a modified Bligh–Dyer method. The extraction was carried using 2:2:2 (*v*/*v*/*v*) water/methanol/dichloromethane at room temperature after spiking internal standards. The organic layer was collected and dried completely under the stream of nitrogen. Dried extract was re-suspended in 100 µL of 10 mM ammonium acetate.

### 4.6. LC-MS/MS

Lipid extract was injected onto a 1.8 μm particle 50 × 2.1 mm id Waters Acquity HSS T3 column (Waters, Milford, MA, USA) heated to 55 °C. The column was eluted with acetonitrile/water (40:60, *v*/*v*) with 10 mM ammonium acetate as solvent A and acetonitrile/water/isopropanol (10:5:85 *v*/*v*) with 10 mM ammonium acetate as solvent B. Gradient is 0 min 40% B, 10 min 100% B, 12 min 100% B, 12.1 min 40% B, and 15 min 40% B. Data were acquired in positive and negative mode using data-dependent MSMS with dynamic mass exclusion at the University of Michigan Metabolomics core. Pooled human plasma sample and pooled experimental sample (prepared by combining small aliquots of all client’s samples) were used to control the quality of sample preparation and analysis. Randomization scheme was used to distribute pooled samples within the set. Mixture of pure authentic standards is used to monitor the instrument performance on a regular basis.

### 4.7. LC-MS/MS Data Analysis

Lipids are identified using LIPIDBLAST package (http://fiehnlab.ucdavis.edu/projects/LipidBlast; accessed on 16 June 2017), computer-generated tandem MS library of 212,516 spectra covering 119,200 compounds representing 26 lipid classes, including phospholipids, glycerolipids, bacterial lipoglycans and plant glycolipids. Quantification of lipids is done by Multiquant software (AB-SCIEX). Data was normalized using internal standards first and cell number. Data from positive and negative ion mode runs were combined with repeats removed and filtered by RSD (<30%).

### 4.8. Lipidomics Data Analysis

Combined normalized data for each run was imported into the R package lipidr (https://github.com/ahmohamed/lipidr; accessed 19 January 2021). Data was imported as a lipidomics object, summarized, annotated, and normalized using the PQN method. Lipidr was used for analysis and to generate graphs.

### 4.9. High-Throughput Lipid Droplet Imaging and Analysis

Huh-7 cells were plated into 96-well optical-bottom plates (Thermo Scientific, 1256670) at the density of 5000 cells/well in conditions similar to other studies. Then, 24 h after oleic acid treatment, cells were fixed in 4% paraformaldehyde for 10 min in RT, washed three times with 1× PBS, and permeabilized with 0.1% Triton-X for 10 min. Intracellular lipid droplets were stained with the LipidTOX Green neutral lipid stain (1:1000; Thermo Scientific, H34475), the nucleus was stained with DAPI (1:1000; Thermo Fisher Scientific, D1306) for 30 min, and cells were then washed twice in PBS. Cells were imaged using the Cellomics Array Scan VT1 (Thermo Fisher Scientific) at a magnification of 20×, and 8 non-overlapping images were captured per well. The images were then imported into the CellProfiler software and processed through automated image analysis pipeline using different analysis modules. Within each image, the pipeline first identifies the nuclei. For each identified nuclei, the pipeline identifies the cell boundary using the propagation algorithm and determines the cell perimeter. Within each identified cell, the program then quantitates lipid droplets’ related variables such as area, diameter, radius, intensity, number of total droplets in cells, etc. For each image, measurements for mean intensity and area of lipid droplets as well as mean lipid droplet number per cell were quantitated. Mean lipid droplet area and mean intensity of droplets represent the average area or intensity for each individual lipid droplet across all image fields for each sample and are independent to the number of cells per field. Combined values across three independent experiments are graphically presented. The pipeline used for CellProfiler imaging is added as Appendix A.

### 4.10. RNA Sequencing and Transcriptome Analysis

Total cellular RNA was prepared from Huh-7 cells using TRIzol reagent (Thermo Fisher Scientific) and purified by ZYMO Clean Kit (Life Technologies) and submitted to the University of Michigan DNA Sequencing Core for quality control, library preparation, and sequencing. RNA sequencing was performed using the Illumina Hi-Seq 4000 platform. The raw sequencing data were 125 base, paired-end reads. Transcriptome analysis was carried out by the University of Michigan bioinformatics core. Briefly, quality of the raw reads data for each sample was checked using FastQC, and the Tuxedo Suite software package was used for alignment and post-analysis diagnostics. FastQC was then used for a second round of quality control (post-alignment), to ensure that only high quality data would be input to expression quantitation and differential expression analysis. Differential gene expression analysis was performed using DESeq2, using a negative binomial generalized linear model (Benjamini–Hochberg FDR (Padj) < 0.05). Genes were annotated with NCBI Entrez Gene IDs and differential expression analysis for lipid metabolizing genes was then performed by using the ipathway Guide (Advaita).

## Figures and Tables

**Figure 1 ijms-22-09758-f001:**
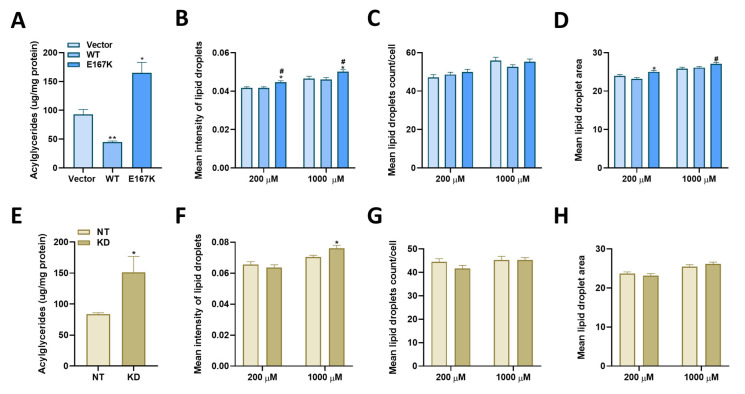
Effects of *TM6SF2* on intracellular lipid accumulation and lipid droplet profile in oleic acid treated Huh-7 cells. (**A**,**E**) Biochemical measurement of intracellular acylglyceride concentrations (TG, DG, MG) in Huh-7 cell lines treated with 200 µM oleic acid for 24 h. Quantitation of (**B**,**F**) mean intensity of lipid droplets, (**C**,**G**) number of lipid droplets per cell and (**D**,**H**) mean area of lipid droplets in Huh-7 cell lines treated by 200 µM or 1000 µM oleic acid for 24 h and stained with Hoechst (nuclei) and LipidTox Green (neutral lipids). Data presented as mean + S.E.M. from 3 independent experiments, * significantly different than control cells (Vector or NT), * *p* < 0.05, ** *p* < 0.01, ^#^ significantly different than wild-type *TM6SF2* overexpression cells (WT), ^#^
*p* < 0.05. *p*-values for (**A**,**E**) were calculated using the Student’s T-test. *p*-values for all other comparisons were calculated by using the Wilcoxon Rank Sum Test. Wild-type *TM6SF2* (WT), E167K mutant variant *TM6SF2* (E167K), pCMV empty vector control (Vector), non-targeted shRNA control (NT), and *TM6SF2* knockdown (KD).

**Figure 2 ijms-22-09758-f002:**
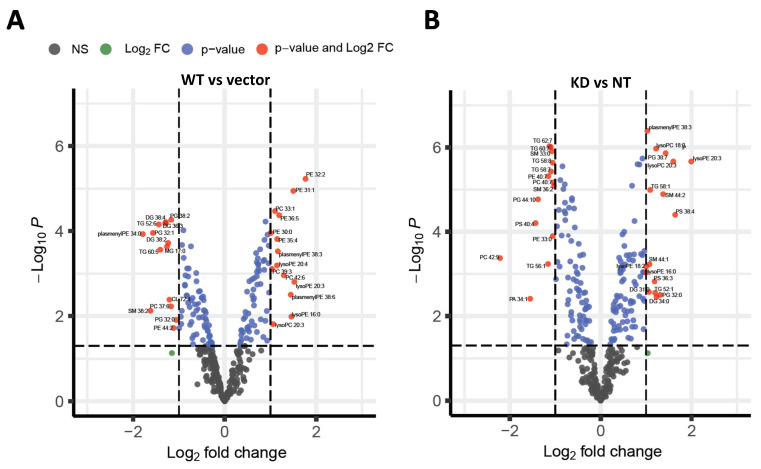
Effects of *TM6SF2* on hepatic lipid profile. Volcano plots showing relative abundance of lipid species identified through untargeted lipidomic analysis. *X*-axis is the log_2_ fold change of lipid species abundance in Huh-7 cells for (**A**) wild-type *TM6SF2* overexpression compared to vector control (**B**) *TM6SF2* knockdown compared to non-targeting control. For both (**A**,**B**), *Y*-axis represents the adjusted *p* value. Negative values indicate downregulated lipids, while positive values upregulated lipids in the overexpression or knockdown cells compared to their respective controls. Vertical dotted lines represent the threshold for log_2_ fold change (>1 or <−1) and horizontal dotted line represent the threshold for adjusted *p* value (Bonferroni corrected; *p* < 0.05). Only the lipid species that met the threshold criteria for both *p*-value and log fold change are labelled (red dots). Values for log fold change and significance for lipids that meet both threshold are shown in detail in Appendix A. Wild-type *TM6SF2* (WT), pCMV empty vector control (Vector), non-targeted shRNA control (NT), and *TM6SF2* knockdown (KD).

**Figure 3 ijms-22-09758-f003:**
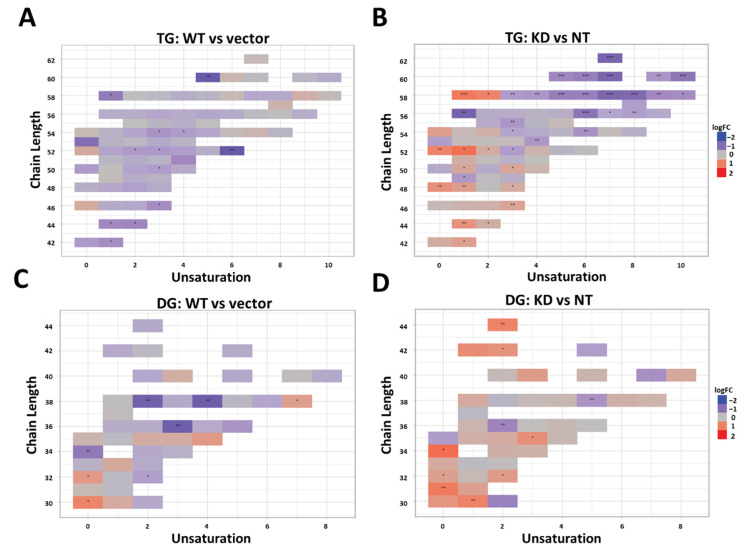
Knockdown of *TM6SF2* decreases longer and more unsaturated TGs and DGs. Heatmap shows the distribution of individual TGs (**A**,**B**) and DGs (**C**,**D**) across total degree of unsaturation and the side chains lengths. Each box represents an individual lipid species where the colors indicate the relative ratio of the lipid (red: increased; white: no difference; blue: decreased) in (**A**,**C**) wild-type overexpression (*n* = 3) compared to vector control (*n* = 3) and (**B**,**D**) *TM6SF2* knockdown (*n* = 5) versus non-targeting control (*n* = 4). Lipids with statistically significant difference between the experimental and control cells are highlighted. * *p* < 0.05, ** *p* < 0.01, *** *p* < 0.001. Wild-type TM6SF2 (WT), pCMV empty vector control (Vector), non-targeted shRNA control (NT), and *TM6SF2* knockdown (KD).

**Figure 4 ijms-22-09758-f004:**
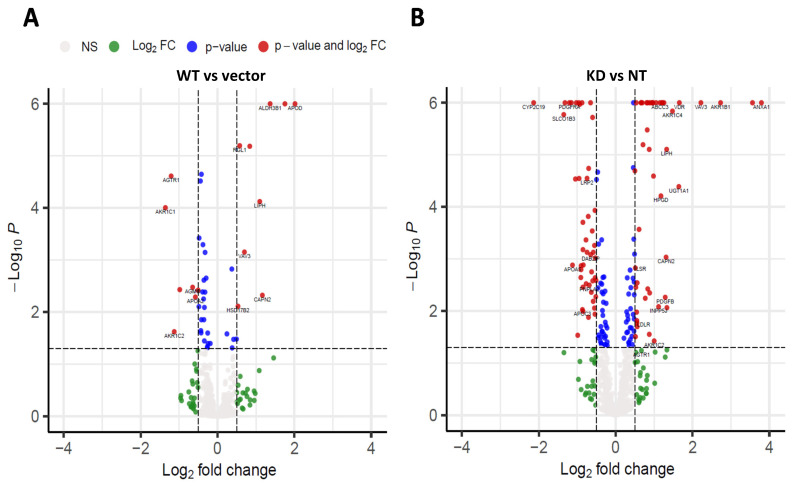
Changes in expression of lipid metabolizing genes in response to *TM6SF2* overexpression or knockout. Volcano plots shows relative abundance of lipid metabolizing genes identified through RNA Sequencing. *X*-axis is the log_2_ fold change of gene expression in Huh-7 cells for (**A**) wild-type overexpression compared to vector control (**B**) *TM6SF2* knockdown compared to non-targeting control. For both (**A**,**B**), *Y*-axis represents the adjusted *p* value. Negative values indicate downregulated genes while positive values indicate upregulated genes in the overexpression or knockdown cells compared to their respective controls. Vertical dotted lines represent the threshold for log_2_ fold change (>0.5 or <−0.5) and horizontal dotted line represent the threshold for adjusted *p* value (Benjamini–Hochberg correction for multiple testing; *p* < 0.05). Genes belonging to the lipase or the AKR1 family of genes that met the threshold criteria for both *p*-value and log fold change are labelled (red dots). Detailed information on the names, fold change and significance of genes meeting both threshold are shown in Appendix A. Wild-type *TM6SF2* (WT), pCMV empty vector control (Vector), non-targeted shRNA control (NT), and *TM6SF2* knockdown (KD).

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
