# Peer review of "Perturbation of TM6SF2 Expression Alters Lipid Metabolism in a Human Liver Cell Line"

_ijms, 2021, doi:10.3390/ijms22189758_

Round 1

Reviewer 1 Report

The manuscript titled as "Perturbation of TM6SF2 expression alters lipid metabolism in a human liver cell line" describes the effects of TM6SF2 expression on lipid profiles in lipid-loaded hepatic cells. The manuscript is very well-written and presents the results soundly. I wish the authors can address a few points below before publication. 1. The effects of E167K mutation in TM6SF2 seem to be dominant negative as the overexpression recapitulate the knock-down phenotypes. Please provide references or comments describing the dominant-negative effects of this mutation. 2. Figure 1: The absolute amount of 'acylglycerides' changes by 2-fold in cells overexpressing WT or E167K, but the magnitude of changes in size or the intensities of lipid probes is very low, in comparison. Please give some comments to explain the differences. 3. Figure 1: the "Mean lipid droplet area" seem to be subjective to cell concentration of the image field. Please define the method more precisely. 4. Figure 1: it is not clear what test the authors use for generating p-values. Please provide the information. 5. Figure 2: it is not clear what "adjusted p-value" means. Please provide information. 6. Figure 2B: it is not clear why total 'acylglyceride' values are increased in knockdown cell-lines but the amounts of TGs in lipidomic analysis have mixed patterns (some TGs are increased and some decreased). Please give some discussion on it. 7. Figure 3: the text/numbers in the figure is too small that it is difficult to read them. 8. Figure 3: please define the chain lengths of "shorter" fatty acids. 9. Figure 4: there is no methods available for the transcriptomic analysis. Please include the methods for the transcriptomic analysis and statistics with it.

Reviewer 2 Report

The manuscript provide an interesting point of view about NAFLD physiopathology. Title is concise, as well keywords. The abstract provide enough information about the manuscript, being complete to the readers.

Attending to the manuscript, introduction is clear, providing a good approach to the issue. Results are well presented, with an appropriate mix of figures and text. Methods are clear, being easy to replicate the study. It is neccessary to provide the Ethical approval for the work (I am not sure if it is cited).

Finally, discussion is clear. However, maybe more cites could be necessary to support the data.

In my personal opinion, the manuscript could be accepted.
